# Does Proton Pump Inhibitors Decrease the Efficacy of Palbociclib and Ribociclib in Patients with Metastatic Breast Cancer?

**DOI:** 10.3390/medicina59030557

**Published:** 2023-03-12

**Authors:** Hatice Odabas, Akif Dogan, Melike Ozcelik, Sedat Yildirim, Ugur Ozkerim, Nedim Turan, Mahmut Emre Yildirim, Mahmut Gumus

**Affiliations:** 1Department of Medical Oncology, University of Health Sciences, Kartal Dr. Lutfi Kirdar City Hospital, Istanbul 34865, Turkey; 2Department of Medical Oncology, University of Health Sciences, Umraniye Training and Research Hospital, Istanbul 34764, Turkey; 3Department of Medical Oncology, Faculty of Medicine, Istanbul Medeniyet University, Istanbul 34722, Turkey

**Keywords:** metastatic breast cancer, palbociclib, ribociclib, proton pump inhibitors, drug-drug interactions

## Abstract

*Background and Objectives:* This investigation aimed to determine the impacts of concurrent proton pump inhibitors (PPIs) on progression-free survival (PFS) in patients with hormone receptor-positive and HER2-negative metastatic breast cancer managed with palbociclib or ribociclib as either the initial or subsequent line of therapy option. *Materials and Methods:* In this retrospective study, patients were classified as “concurrent PPIs” if PPIs were given for at least two-thirds of the palbociclib or ribociclib therapy period, and “no concurrent PPIs” if no PPIs were given during the period of palbociclib or ribociclib therapy. Each patient was also classified as endocrine-sensitive or endocrine-resistant according to the duration of previous endocrine responses. “Concurrent PPIs” and “no concurrent PPIs” groups were compared with each other in terms of PFS. This comparison was performed for both ribociclib and palbociclib groups. *Results*: The research included 220 patients in total. The PFS of 57 patients on palbociclib using concomitant PPIs was 14.4 months. Among 63 patients using palbociclib without concomitant PPIs, the PFS was 15.8 months. No statistically significant difference was found with PPI use (*p* = 0.82). Among 29 patients using ribociclib concurrently with PPIs, the PFS was 22.4 months. Among 71 patients using ribociclib without PPIs, the PFS was 20.2 months. No statistically significant difference was found with PPI use (*p* = 0.40). *Conclusion*: The results of our investigation showed that concomitant use of the most commonly used PPIs in the study (lansoprazole, pantoprazole, and esomeprazole) with palbociclib or ribociclib did not have any detrimental effects on PFS. Where appropriate, PPIs can be used concurrently with palbociclib and ribociclib. However, the effect of PPIs on cycling-dependent kinase 4/6 inhibitors deserves further investigation.

## 1. Introduction

Drug–drug interactions (DDIs) are a serious issue in oncology, particularly in the current era of extensive oral treatments [1]. The absorption phase is a significant step in drug metabolism. During the process of absorption, drug interaction may affect bioavailability. The most common interactions influencing bioavailability are those caused by changes in gastric pH, usually due to the use of proton pump inhibitors (PPIs), antacids, and H2-receptor antagonists [2]. DDIs, especially at the absorption level, should be regarded as a probable explanation of cancer therapy failure [3]. For this reason, concern about the concomitant use of PPIs with tyrosine kinase inhibitors that have pH-dependent solubility is considered here [4,5]. While concurrent use of pazopanib with PPIs has been shown to decrease the anti-tumoral effect of pazopanib, it has been observed in other studies that PPIs do not affect clinical outcomes [6,7]. Similarly, in a pooled analysis of phase II and phase III studies of sunitinib, axitinib, and sorafenib, it was reported that concomitant use of different PPIs has no unfavorable impacts on the efficacy and safety of these drugs [8].

Patients with breast cancer have been identified to have hyperactive cyclin-dependent kinase (CDK) 4/6 pathways. Oral CDK 4/6 inhibitors (CDK 4/6i), such as palbociclib and ribociclib, have been demonstrated to have therapeutic efficacy in conjunction with aromatase inhibitors or fulvestrant as the first- or second-line therapies of premenopausal and postmenopausal patients with HER2-negative metastatic and hormone receptor-positive breast cancer (MBC) [9,10,11,12]. The palbociclib free base capsule is a weak base with pH-dependent solubility. The palbociclib free base capsule completely dissolves at pH < 4.3 in 250 mL media. Gastric pH increases to above 4.5 with the use of PPIs, which has been shown to decrease the solubility of palbociclib. This study evaluated the potential impact of PPI (rabeprazole) on the absorption and pharmacokinetics of palbociclib in healthy volunteers.

Concurrent administration of palbociclib and rabeprazole at fasting decreases the mean area under the concentration-time curve from time 0 to infinity (AUCinf) of palbociclib by 62% and the maximum observed plasma concentration (Cmax) of palbociclib by 80%. After a meal, these values were a 13% decrease in AUCinf and a 41% decrease in Cmax. Given the clinically insignificant effects after a meal, the product label did not limit the concurrent use of palbociclib, and PPI [13,14].

Ribociclib is also a weak base with pH-dependent solubility. While ribociclib solubility is expected to decrease with an increase in gastric pH, changes in gastric pH were not shown to affect the solubility and pharmacokinetics of ribociclib [15,16]. Concurrent use of ribociclib with PPIs and other gastric pH-increasing drugs was not limited per the label [17,18]. The effect of concurrently-used CDK 4/6i with PPIs on progression-free survival (PFS) time was evaluated in two retrospective trials. In contrast with the results of pharmacokinetics trials of palbociclib and ribociclib, detrimental effects on PFS time were observed in MBC patients [19,20].

Due to the uncertain and limited data on the effect of concurrent use of CDK 4/6i and PPIs on clinical outcomes, the present investigation aimed to examine the impact of the use of concomitant PPIs on the PFS in HER2-negative and hormone receptor-positive MBC patients who were treated with palbociclib or ribociclib as either the initial or subsequent line of therapy.

## 2. Material and Methods

This retrospective analysis included HER2-negative and hormone receptor-positive MBC patients treated with palbociclib or ribociclib with or without concurrent PPIs. HER2-negative, hormone receptor-positive breast cancers were classified as tumors with more than 10% estrogen receptor activity and a negative HER2 test result (score 0 or 1+ or negative with immunohistochemistry and negative staining with dual-probe in situ hybridization).

Patients were classified as “concurrent PPIs” if PPIs were given for at least two-thirds of the palbociclib or ribociclib therapy period, and “no concurrent PPIs” if no PPIs nor antacids and H2 antagonists were given during palbociclib or ribociclib therapy. Patients using cytochrome P450 3A4 (CYP3A4) antagonists or enhancers were not allowed to be included in the study. Additionally, patients had to receive PPIs, such as omeprazole 40 mg, pantoprazole 40 mg, rabeprazole 20 mg, esomeprazole 40 mg, and lansoprazole 30 mg, with breakfast in order to be eligible. Each patient was classified as endocrine-sensitive or endocrine-resistant according to the duration of previous endocrine responses. Endocrine sensitivity was defined as “relapse” for at least 12 months of relapse after the completion of adjuvant endocrine therapy or with de novo advanced breast cancer. Endocrine resistance was defined as relapse after the first 2 years while on adjuvant endocrine therapy, relapse within 12 months of completing adjuvant endocrine therapy, or progressive disease (PD) ≥ 6 months after initiating endocrine therapy for MBC, while on endocrine therapy [21].

Palbociclib 125 mg (capsule form) and ribociclib 600 mg were administered once daily for three weeks, pursued by 7 days off, which was repeated every 28 days (in conjunction with fulvestrant or continuous aromatase inhibitor treatment [and a gonadotropin-releasing hormone agonist if pre- or perimenopausal female]); therapy was continued until disease progression or intolerable toxic effects. Palbociclib was decreased to 100 mg/day or 75 mg/d, while ribociclib was adjusted to 400 mg/day or 200 mg/d based on tolerability. 

“Concurrent PPIs” and “no concurrent PPIs” groups were compared regarding PFS. This comparison was performed for both ribociclib and palbociclib groups. Tumor response was assessed using Response Evaluation Criteria in Solid Tumors (RECIST) (version 1.1) every 12 weeks until disease progression, death, or loss of follow-up; for patients who discontinued for any other reason [22]. Toxicity was graded according to Common Terminology Criteria for Adverse Events (CTCAE v5).

The study was conducted in accordance with the Helsinki Declaration Principles, and ethics committee approval was obtained from the Ethics Committee of Kartal Dr. Lutfi Kirdar City Hospital.

## 3. Statistical Analysis

“Progression free survival time” was considered the primary hypothesis to be investigated in the study. The sample size was calculated according to the expected distribution of different treatment options, when calculating at a 95% level of confidence and 5% type 1 error levels. Assuming a pooled standard deviation of 9 units, the study would require a sample size of 51 for each group (i.e., a total sample size of 102, assuming equal group sizes) to achieve a power of 80% and a level of significance of 5% (two-sided), for detecting an actual difference in means between the test and the reference group of 5 months [23].

The absolute and relative frequencies were used to represent the categorical data, whereas the median and range were used to depict the quantitative variables. Either chi-square or Fisher’s exact tests were employed to assess the clinical and pathological characteristics of the patients. The PFS was defined as the period that lasted from the initiation of therapy to the advancement of the illness, loss to follow-up, or date of death. The Kaplan–Meier modality was used to calculate the survival curves, which were then analyzed using the log-rank test. The Cox hazard regression modality was utilized to find the independent factors associated with PFS. The overall survival (OS) was defined as the period that lasted from the initiation of therapy to the date of death. All *p* values were two-sided, and *p* < 0.05 was considered statistically significant. SPSS 16.0 (SPSS Inc., Chicago, IL, USA) was used to analyze all data.

## 4. Results

In the study, 220 patients were included between 22 May 2020 and 13 April 2022. The median age was 57 (range = 25–92) years. Of the patients, 49.1% had visceral disease and 75% were postmenopausal. Of these 220 patients, 120 received palbociclib and 100 received ribociclib (Figure 1). Table 1 and Table 2 illustrate the clinical and treatment features of the patients, as well as the types of PPIs utilized.

### 4.1. The Palbociclib Group

In the palbociclib group, 63 patients were not administered any PPIs during palbociclib treatment, whereas 57 patients were on concurrent palbociclib–PPI medication. Palbociclib dosages were reduced in 31.7% of the patients (Table 2). Neutropenia and leukopenia were the most commonly observed grade 3–4 adverse events (AEs) (Table 3).

The overall response rate (ORR) was 63.3%, with an 83.3% disease control rate (DCR) (Table 4). There was no substantial difference in the palbociclib group regarding AEs, ORR, or DCR between patients who received PPIs and those who did not (*p* = 0.44, *p* = 0.24, and *p* = 0.80, respectively).

The median follow-up time was 14.1 months. In the course of therapy, the progression of the disease was observed in 63 patients (52.5%). The median PFS time was 15.7 (95% confidence interval [CI], 13.2–18.3) months. The univariate analysis included the following variables: age, Eastern Cooperative Oncology Group performance status (ECOG PS), menopausal status, visceral disease, bone metastasis, CDK4/6i type, number of metastatic sites, endocrine sensitivity or resistance status, dose reduction, and concomitant use of PPIs (Table 5). 

The median PFS time for 59 patients with visceral metastatic disease was 13.7 (95% CI, 9.5–17.8) months. The median PFS duration was 16.1 months in 61 patients without the visceral metastatic disease (hazard ratio [HR], 1.40; 95% CI, 0.84–2.34; *p* = 0.19). Among the 57 patients using concomitant PPIs, the median PFS time was 14.4 (95% CI, 8.1–20.6) months. Among 63 patients using palbociclib without concomitant PPIs, the median PFS time was 15.8 (95% CI, 13.6–17.9) months. No statistical significance was found for PPI use (HR, 1.07; 95% CI, 0.64–1.75; *p* = 0.82) (Figure 2). 

Patients with endocrine-sensitive tumors had a median PFS time of 9.5 months longer than those with endocrine-resistant tumors (19.7 months vs. 10.2 months). Only endocrine sensitivity or resistance was observed to strongly influence the median PFS time (HR, 2.53; 95% CI, 1.46–4.39; *p* = 0.001). Endocrine resistance was established as the primary independent predictive marker for reduced PFS time (HR, 0.37; 95% CI, 0.20–0.67; *p* = 0.001). The multivariate analysis indicated that concurrent PPI usage was not a predictive biomarker for shorter PFS times (HR, 0.98; 95% CI, 0.59–1.65; *p* = 0.95) (Table 5).

The PFS analysis evaluated the role of PPIs in endocrine sensitivity by dividing the patients into four groups according to their endocrine sensitivity and PPI usage status. Among the endocrine-sensitive patients, the median PFS time in the “no concomitant PPIs” group (*n* = 24) was 19.7 months (95% CI, 12.4–27.0), while it was not reached in the “concomitant PPIs” group (*n* = 30) (*p* = 0.88). Among the endocrine-resistant patients, the median PFS time in the “no concomitant PPIs” group (*n* = 39) was 10.45 months (95% CI, 4.65–16.25), while it was 9.2 months (95% CI, 2.5–15.9) in the “concomitant PPIs” group (*n* = 27) (*p* = 0.71). The data were too immature to estimate the median OS because throughout the follow-up period, 24.2% (*n* = 29) of patients in the palbociclib group died. 

### 4.2. The Ribociclib Group

In the ribociclib group, 71 patients received no PPIs during ribociclib therapy, whereas 29 patients received ribociclib–PPI treatment concurrently. In 37% of the patients, the ribociclib dosage was reduced (Table 2). The most commonly observed grade 3–4 AEs were neutropenia and leukopenia (Table 3). In the ribociclib group, the ORR was 69% and the DCR was 89.7% (Table 4). Adverse events, ORR, and DCR, were similar between patients who were given PPIs and those who were not (*p* = 0.64, *p* = 0.24, and *p* = 0.40, respectively).

The median follow-up time was 11.9 months. Forty-one percent of patients experienced progressive disease during the follow-up period. The median PFS time was 20.3 (95% CI, 16.0–24.2) months. Age, ECOG PS, menopausal status, visceral illness, bone metastasis, CDK4/6i type, number of metastatic locations, endocrine sensitivity or resistance status, dosage decrease, and concurrent usage of PPIs were all considered in the univariate analysis (Table 5).

The median PFS time was 18.7 months (95% CI, 7.1–30.3) among the 49 patients diagnosed with visceral metastatic cancer and 20.5 months (95% CI, 15.8–25.1) (HR, 0.58; 95% CI, 0.31–1.09; *p* = 0.09) among the 51 patients without visceral metastasis. The median PFS time was 22.4 months (95% CI, 9.0–35.9) among the 29 patients using concurrent PPIs. The median PFS time among the 71 patients without concurrent PPIs was 20.2 months (95% CI, 17.1–23.4). PPI usage had no statistically significant effect on PFS (HR, 1.36; 95% CI, 0.66–2.77; *p* = 0.40) (Figure 3).

The median PFS time in patients sensitive to earlier endocrine treatment could not be determined. The median PFS time in patients who had previously failed endocrine therapy was 17.5 months (95% CI, 10.2–24.8). Only endocrine sensitivity or resistance status significantly influenced PFS time (HR, 2.91; 95%CI, 1.29–6.58; *p* = 0.01). The existence of endocrine resistance was identified as the main independent predictive parameter related to shorter PFS time in the multivariate analysis (HR, 0.28; 95%CI, 0.12–0.66; *p* = 0.004). The multivariate analysis indicated that concurrent PPI usage was not significantly related to PFS time (HR,1.71; 95% CI, 0.78–3.78; *p* = 0.18) (Table 5).

In the PFS analysis, among the endocrine-sensitive patients, seven of the patients in the ‘’no concomitant PPIs’’ group had disease progression. In contrast, none of the patients in the ‘’concomitant PPIs’’ group had disease progression. Therefore, PFS analysis was not done. 

Among the endocrine-resistant patients, the median PFS time in the ‘’no concomitant PPIs’’ group was 17.5 months (95% CI, 8.9–26.1) while it was 14.7 months (95% CI, 7.2–24.8) in the ‘’concomitant PPIs’’ group, (*p* = 0.81). The data were immature to estimate the median OS because 17% (*n* = 17) of patients in the ribociclib group were dead throughout the follow-up period. 

## 5. Discussion

Proton pump inhibitor-induced increases in gastric pH have been found to reduce the absorption and exposure of weak base medications with pH-dependent solubility [24]. To our knowledge, this is the first study to indicate that PPIs (those most commonly used in the study) administered concurrently with palbociclib or ribociclib have no negative impact on PFS in MBC patients.

After a meal, rabeprazole had no clinically significant effect on the Cmax (41% decrease) or AUCinf (13% decrease) of palbociclib [13,14]. Samant et al. concluded that food intake and PPIs do not clinically affect the PK of ribociclib [15]. Lu et al. evaluated the population PK (popPK) of ribociclib [16]. In the popPK analysis, age, sex, race, ECOG PS 1, mild hepatic impairment, mild or moderate renal impairment, and concomitant use of letrozole, anastrozole, fulvestrant, PPIs, or weak CYP3A4/5 inhibitors had no impact on the PK of ribociclib. 

In the current study, we revealed that concurrent usage of PPIs with palbociclib (capsule form) and ribociclib did not have a detrimental effect on PFS (14.4 months vs. 15.8 months, *p* = 0.82; 22.4 months vs. 20.2 months, *p* = 0.40). This finding was compatible with the results of PK trials of palbociclib and ribociclib [13,15,16]. Palbociclib and ribociclib are metabolized mainly by CYP3A4, and rabeprazole is metabolized via the nonenzymatic pathway [14,17,18,25]. The difference among the elimination pathways indicates no clinically significant metabolic interaction between palbociclib and PPIs. The results of our study support this idea. Furthermore, the response rates and grade 3–4 toxicities among the patients using palbociclib and ribociclib with or without PPIs were similar. Thus, the PPIs did not affect the bioavailability of palbociclib and ribociclib. 

A palbociclib and letrozole combination was approved as the first-line treatment of patients with endocrine-sensitive MBC in the Paloma-2 trial [9]. In this trial, the PFS time was 24.8 months. All the patients were postmenopausal and chemotherapy-naive for metastatic disease. In the current study, the PFS time was 19.7 months with first-line letrozole and palbociclib in the endocrine-sensitive group. Our study included premenopausal patients (16.7%) and patients with a history of chemotherapy (20.4%). In addition, our study had a shorter follow-up period (13.8 months). These features likely account for the shorter PFS in our study compared to that in the Paloma-2 trial.

The addition of palbociclib to fulvestrant increased the PFS in the Paloma-3 trial (9.5 months vs. 4.6 months) [26]. The PFS of our patients in the endocrine-resistant group was 10.2 months, similar to the findings of Paloma-3.

In Moonaleesa-2, adding ribociclib to letrozole as a first-line treatment improved PFS during 26 months of follow-up (25.3 months vs. 16.0 months) [11]. The number of endocrine-sensitive patients in the ribociclib group was limited in our study (*n* = 34). There was progressive disease in only seven patients, and median PFS was not reached.

According to the results of Monaleesa-3, the addition of ribociclib to fulvestrant improved PFS (20.5 months vs. 12.8 months) [12]. The median PFS of the patients in the endocrine-resistant group was 17.5 months. Monaleesa-3 included endocrine-sensitive patients (49.2%), and endocrine-resistant patients. The patients who used previous chemotherapy at the metastatic stage were not included. All the patients in this group were endocrine-resistant in our study. About 30% of patients were premenopausal and 45.9% had a history of chemotherapy for metastatic disease. Some patients used ribociclib as a third or later-line treatment (19.1%) and an aromatase inhibitor as concurrent endocrine therapy (27.9%). These features impacted the shorter PFS we obtained compared to Monaleesa-3. In summary, the results of endocrine-sensitive and endocrine-resistant patients in our study were compatible with the literature when the limited patient number and the difference in patient characteristics were considered [9,10,11,12]. Furthermore, in endocrine-sensitive and endocrine-resistant patient groups, we did not observe a negative impact of concurrent PPIs on PFS (*p* > 0.05 for each).

Although no unfavorable effect of PPI use on palbociclib or ribociclib treatment is expected, Del Re et al. and Eser et al. demonstrated that concomitant use of palbociclib with PPIs had a detrimental effect on PFS in patients with MBC treated with palbociclib (PFS time 14.0 months vs. 37.9 months, *p* < 0.0001 and 13.0 months vs. not-reachable, *p* < 0.0001, respectively) [19,20]. In their study, Eser et al. concluded that the PFS of patients using PPIs and ribociclib was shorter than that of patients not using PPIs (12.6 months vs. not-reachable, *p* = 0.003) (20). Del Re et al. and Eser et al. reported that gastric pH increases with PPI use, and the efficacy of palbociclib might decrease by the reduction in plasma concentration.

However, in PK trials with concurrent PPI use, there was no decrease in the efficacy of palbociclib free base capsule in the satiety state and ribociclib in a fasting or satiety state [13,15,16]. The results of these two retrospective studies were not compatible with the findings of PK trials. The PFS advantage with palbociclib was 10.3 months in Paloma-2 (24.8 months vs. 14.5 months) and 6.6 months (11.2 months vs. 4.6 months) in Paloma-3 [9,10]. Moreover, there was a 23-month median PFS difference (37.9 months vs. 14.0 months) between patients with and without concurrent PPI use in Del Re et al.’s trial. These findings were expected to be at least similar to the results of placebo-receiving patients. Even though PPIs may have a detrimental effect through a different mechanism, preclinical studies do not support this event. On the contrary, preclinical data suggest that PPIs may have anti-proliferative effects against breast cancer cells [27]. Possible explanations for the shorter PFS of patients using concomitant PPIs could be various, including tumor burden and non-registered drugs in use.

There were some limitations to our study. A moderate sample size and retrospective nature are among these limitations. In addition, the inability to perform OS analysis due to the lack of sufficient events during the follow-up period was one of the limitations of the study. Although PFS is a valuable outcome in managing patients with MBC in oncology practice, the negative impact of confounding variables on the performance of PFS due to the study’s retrospective design was a limitation. The other limitation is related to the unequal use of PPIs in number. Lansoprazole, pantoprazole, and esomeprazole were the most commonly used PPIs in our study. PPIs may differ from each other in terms of potency. Receipt of different PPIs has the potential to interfere differently with the efficacy of the concurrently used treatments. Again, the results of the present study may only be generalized for some PPIs. Omeprazole was the least frequently used PPI in this study, so findings may not represent patients using omeprazole. An exploratory analysis of the recent phase II PARCIFAL trial [28] and future trials may help to assess if the results of this study can be replicated in patient populations using predominantly omeprazole. 

This research provides important data on the concomitant use of the most commonly used PPIs in the study with palbociclib or ribociclib, which does not negatively affect PFS. These PPIs and CDK4/6i can be used concurrently for appropriate indications. Nevertheless, the impact of PPIs on CDK4/6i warrants further research.

## Figures and Tables

**Figure 1 medicina-59-00557-f001:**
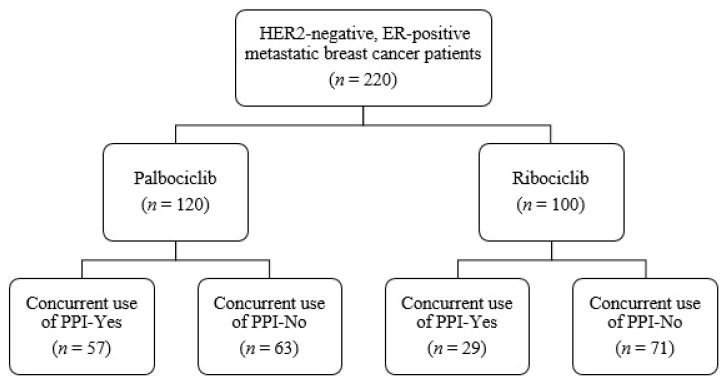
Flow chart of the study participants. Abbreviations: ER, estrogen receptor; HER2, human epidermal growth factor receptor 2; PPI, proton pump inhibitor.

**Figure 2 medicina-59-00557-f002:**
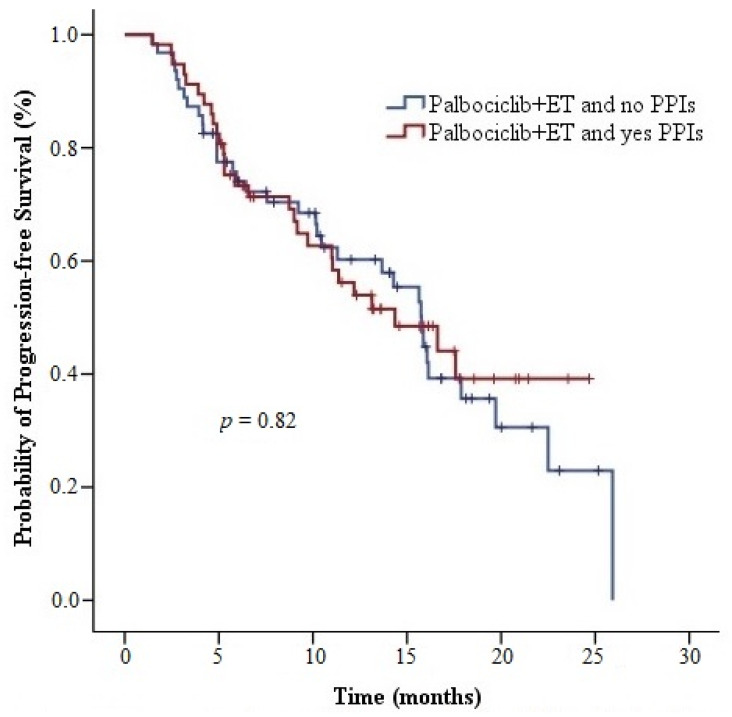
Progression-free survival curves of palbociclib + endocrine therapy with or without proton pump inhibitor. Abbreviations: ER, estrogen receptor; PPI, proton pump inhibitor.

**Figure 3 medicina-59-00557-f003:**
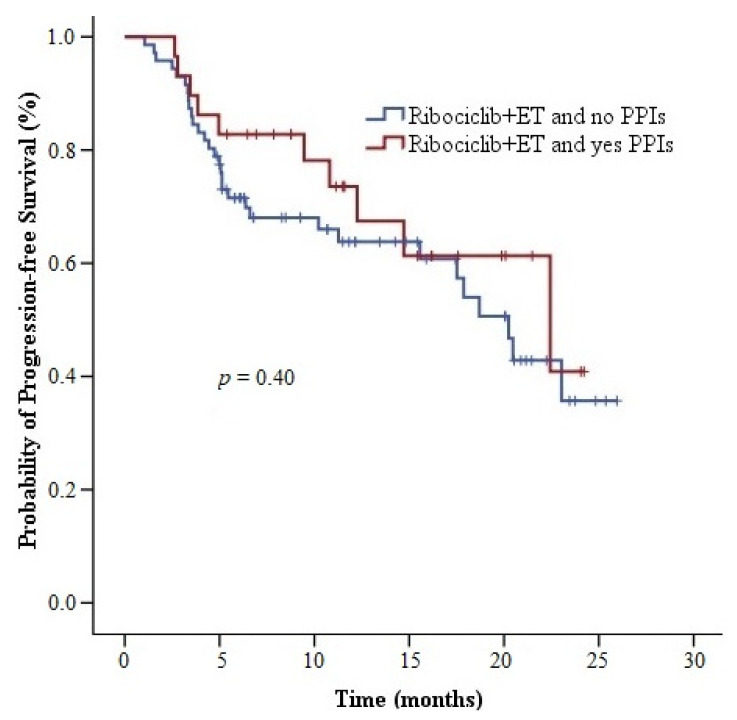
Progression-free survival curves of ribociclib + endocrine therapy with or without proton pump inhibitor. Abbreviations: ER, estrogen receptor; PPI, proton pump inhibitor.

**Table 1 medicina-59-00557-t001:** Clinical characteristics of patients and their distribution across PPI groups.

Characteristics	Palbociclib				Ribociclib			
	Total Number of Patients (*n* = 120)	Concurrent Use of PPIs		P Score	Total Number of Patients (*n* = 100)	Concurrent Use of PPIs		P Score
		Yes (*n* = 57)	No (*n* = 63)			Yes (*n* = 29)	No (*n* = 71)	
Age (years), median (range)	58 (25–92)	60 (33–92)	54 (25–86)	0.07	56 (31–84)	60 (34–84)	53 (31–80)	0.17
**Menopausal status, *n (*%) ***								0.16
Premenopause	27 (22.7)	8 (14.0)	19 (30.6)	0.03	27 (27.0)	5 (17.2)	22 (31.0)	
Postmenopause	92 (77.3)	49 (86.0)	43 (69.4)		73 (73.0)	24 (82.8)	49 (69.0)	
**ECOG PS, *n* (%)**				0.58				0.44
0	68 (56.7)	33 (57.9)	35 (55.6)		46 (46.0)	16 (55.2)	30 (42.3)	
1	40 (33.3)	20 (35.1)	20 (31.7)		48 (48.0)	11 (37.9)	37 (52.1)	
2–3	12 (10.0)	4 (7.0)	8 (12.7)		6 (6.0)	2 (6.9)	4 (5.6)	
**Disease site *n* (%)**				0.72				0.43
Visceral †	59 (49.2)	29 (50.9)	30 (47.6)		49 (49.0)	16 (55.2)	33 (46.5)	
Non-visceral	61 (50.8)	28 (49.1)	33 (52.4)		51 (51.0)	13 (44.8)	38 (53.5)	
**Bone lesion only *n (*%)**				0.70				0.16
Yes	37 (30.8)	19 (33.3)	18 (28.6)		27 (27.0)	5 (17.2)	22 (31.0)	
No	83 (69.2)	38 (66.7)	45 (71.4)		73 (73.0)	24 (82.8)	49 (69.0)	
**Visceral *n (*%)**								
Lung	38 (31.7)	16 (28.1)	22 (34.9)	0.42	18 (18.0)	13 (44.8)	18 (25.4)	0.56
Liver	24 (20.0)	12 (21.1)	12 (19.0)	0.78	31 (31.0)	2 (6.9)	16 (22.5)	0.65
Lung + liver	52 (43.3)	25 (43.9)	27 (42.9)	0.91	41 (41.0)	13 (44.8)	28 (39.4)	0.62
**Number of metastatic site,** ***n* (%)**				0.44				0.22
<3	80 (66.7)	40 (70.2)	40 (63.5)		51 (51.0)	12 (41.4)	39 (54.9)	
≥3	40 (33.3)	17 (29.8)	23 (36.5)		49 (49.0)	17 (58.6)	32 (45.1)	

Abbreviation: ECOG PS, Eastern Cooperative Oncology Group (ECOG) performance status; PPI, proton pump inhibitor. * One patient in palbociclib group is male. † Visceral metastasis was defined as lung, liver, brain, pleural, or peritoneal involvement.

**Table 2 medicina-59-00557-t002:** Treatment characteristics of patients and their distribution across PPI groups.

Characteristics	Palbociclib				Ribociclib			
	Total Number of Patients (*n* = 120)	Concurrent Use of PPIs		P Score	Total Number of Patients (*n* = 100)	Concurrent Use of PPIs		P Score
		Yes (*n* = 57)	No (*n* = 63)			Yes (*n* = 29)	No (*n* = 71)	
**Endocrine-sensitive or resistant disease, *n* (%)**				0.11				0.90
Sensitive	54 (45.0)	30 (52.6)	24 (38.1)		32 (32.0)	9 (31.0)	23 (32.4)	
Resistant	66 (55.0)	27 (47.4)	39 (61.9)		68 (68.0)	20 (69.0)	48 (67.6)	
**CDKI *n* (%)**				0.33				0.59
1. line setting	70 (58.3)	37 (64.9)	33 (52.4)		52 (52.0)	13 (44.8)	39 (54.9)	
2. line setting	37 (30.8)	14 (24.6)	23 (36.5)		35 (35.0)	11 (37.9)	24 (33.8)	
≥3. line setting	13 (10.8)	6 (10.5)	7 (11.1)		13 (13.0)	5 (17.2)	8 (11.3)	
**Previous chemotherapy *n (*%)**				0.36				0.70
Neoadjuvant/adjuvant treatmentonly	39 (32.5)	23 (40.4)	16 (25.4		35 (35.0)	11 (37.9)	24 (33.8)	
Metastatic treatment only	21 (17.5)	8 (14.0)	13 (20.6)		25 (25.0)	5 (17.2)	20 (28.2)	
Neoadjuvant/ adjuvant + Metastatic treatment	18 (15.0)	8 (14.0)	10 (15.9)		14 (14.0)	5 (17.2)	9 (12.7)	
None	42 (35.0)	18 (31.6)	24 (38.1)		26 (26.0)	8 (27.6)	18 (25.4)	
**CDKI + ET *n* (%)**				0.86				0.33
Letrazole	79 (65.8)	38 (66.7)	41 (65.1)		46 (46.0)	11 (37.9)	35 (49.3)	
Fulvestrant	41 (34.2)	19 (33.3)	22 (34.9)		52 (52.0)	18 (62.1)	34 (47.9)	
Other	0 (0.0)	0 (0.0)	0 (0.0)		2 (2.0)	0 (0.0)	2 (2.8)	
**Dose reduction of CDKI *n* (%)**				0.44				0.30
Yes	38 (31.7)	20 (35.1)	18 (28.6)		37 (37.0)	13 (44.8)	24 (33.8)	
No	82 (68.3)	37 (64.9)	45 (71.4)		63 (63.0)	16 (55.2)	46 (64.8)	
**Dose interruption of CDKI *n (*%)**				0.82				0.53
Yes	37 (30.8)	17 (29.8)	20 (31.7)		40 (40.0)	13 (44.8)	27 (38.0)	
No	83 (69.2)	40 (70.2)	43 (68.3)		60 (60.0)	16 (55.2)	44 (62.0)	
**PPI, *n* (%)**								
Lansoprazole		17 (29.8)				14 (48.3)		
Pantoprazole		14 (24.6)				7 (24.1)		
Esomeprazole		16 (28.1)				4 (13.8)		
Rabeprazole		5 (8.8)				3 (10.3)		
Omeprazole		5 (8.8)				1 (3.4)		

Abbreviation: CDKI, cyclin dependent kinase 4/6 inhibitors (palbociclib or ribociclib); ET, endocrine therapy; PPIs, proton pump inhibitors.

**Table 3 medicina-59-00557-t003:** Grade 3–4 adverse events.

AE	Palbociclib				Ribociclib			
	Total Number of Patients (*n* = 120)	Concurrent Use of PPIs		P Score	Total Number of Patients (*n* = 100)	Concurrent Use of PPIs		P Score
		Yes (*n* = 57)	No (*n* = 63)			Yes (*n* = 29)	No (*n* = 71)	
Any AE	40 (33.3)	17 (29.8)	23 (36.5)	0.44	45 (45.0)	12 (41.4)	33 (46.5)	0.64
Neutropenia *	33 (27.5)	15 (26.3)	18 (28.6)	0.80	37 (37.0)	11 (37.9)	26 (36.6)	0.90
Leukopenia †	18 (15.0)	9 (15.8)	9 (14.3)	0.82	19 (19)	6 (20.7)	13 (18.3)	0.78
Anemia ‡	6 (5.0)	2 (3.5)	4 (6.3)	0.70	3 (3.0)	1 (3.4)	2 (2.8)	0.87
Thrombocytopenia §	2 (1.7)	0 (0.0)	2 (3.2)	0.50	3 (3.0)	1 (3.4)	2 (2.8)	0.87
Abnormal LFTs ǁ	1 (0.8)	0 (0.0)	1 (1.6)		4 (4.0)	0 (0.0)	4 (5.6)	0.32
Fatigue	3 (2.5)	1 (1.8)	2 (3.2)	1.00	1 (1.0)	0 (0.0)	1 (1.4)	0.52
Decreasedappetite	1 (0.8)	0 (0.0)	1 (1.6)	1.00	0 (0.0)	0 (0.0)	0 (0.0)	
Nause	0 (0.0)	0 (0.0)	0 (0.0)		2 (2.0)	1 (3.4)	1 (1.4)	0.50
Vomiting	0 (0.0)	0 (0.0)	0 (0.0)		2 (2.0)	1 (3.4)	1 (1.4)	0.50
Stomatitis	1 (0.8)	0 (0.0)	1 (1.6)	1.00	0 (0.0)	0 (0.0)	0 (0.0)	
Thromboembolic events	1 (0.8)	1 (1.8)	0 (0.0)	0.48	0 (0.0)	0 (0.0)	0 (0.0)	
Pulmonary toxicity	1 (0.8)	0 (0.0)	1 (1.6)	1.00	1 (1.0)	0 (0.0)	1 (1.4)	1.00
Prolonged QT interval	0 (0.0)	0 (0.0)	0 (0.0)		3 (3.0)	1 (3.4)	2 (2.8)	

Abbreviation: AE, adverse event; LFT, liver function test; PPI, proton pump inhibitor. * Neutropenia includes neutropenia, decreased neutrophil count, febrile neutropenia, and neutropenic sepsis. † Leukopenia includes leukopenia, decreased white blood cell count, lymphopenia, and decreased lymphocyte count. ‡ Anemia includes anemia, decreased hemoglobin level, and decreased red blood cell count. § Trombocytopenia includes decreased platelet count and thrombocytopenia. ǁ Abnormal LFT includes increased alanine amino transferase, increased aspartate amino transferase, and increased blood bilirubin.

**Table 4 medicina-59-00557-t004:** Overall tumor response.

	**Palbociclib**				Ribociclib			
Tumor Response	Total Number of Patients (*n* = 120)	Concurrent Use of PPIs		P Score	Total Number of Patients (*n* = 100)	Concurrent Use of PPIs		P Score
		Yes (*n* = 57)	No (*n* = 63)			Yes (*n* = 29)	No (*n* = 71)	
CR	4 (3.3)	3 (5.3)	1 (1.6)	0.35	1 (1.0)	1 (3.4)	0 (0.0)	0.39
PR	72 (60)	30 (52.6)	42 (66.7)	0.12	59 (59.0)	19 (65.5)	40 (56.3)	0.40
ORR *	76 (63.3)	33 (57.9)	43 (68.3)	0.24	60 (60.0)	20 (69.0)	40 (56.3)	0.24
SD	24 (20.0)	14 (24.6)	10 (15.9)	0.23	24 (24.0)	6 (20.7)	18 (25.4)	0.62
DCR †	100 (83.3)	47 (82.5)	53 (84.1)	0.80	84 (84.0)	26 (89.7)	58 (81.7)	0.40
PD	20 (16.7)	10 (17.5)	10 (15.9)	0.80	16 (16.0)	3 (10.3)	13 (18.3)	0.40

Abbreviation: CR, complete response; PR, partial response; ORR, overall response rate; SD, stable disease; DCR, disease control rate; PD, progressive disease; PPI, proton pump inhibitor. * Overall response included CR or PR. † Disease control was defined as CR or PR, SD ≥ 24 weeks, or neither CR nor PD ≥ 24 weeks.

**Table 5 medicina-59-00557-t005:** Univariate and multivariate analysis for progression-free survival.

Subgroup	Palbociclib				Ribociclib			
	Univariate		Multivariate		Univariate		Multivariate	
	HR (95%CI)	P Score	HR (95%CI)	P Score	HR (95%CI)	P Score	HR (95%CI)	P Score
Age	1.51 (0.91–2.52)	0.11	1.68 (0.83–3.39)	0.15	0.67 (0.31–1.47)	0.32	1.99 (0.81–4.94)	0.13
ECOG PS(1)	0.78 (0.45–1.37)	0.40	0.55 (0.28–1.09)	0.09	0.72 (0.36–1.45)	0.36	0.82 (0.38–1.79)	0.62
Post/Pre-menopause	1.41 (0.75–2.65)	0.29	0.80 (0.39–1.67)	0.56	0.77 (0.40–1.50)	0.45	0.84 (0.41–1.72)	0.64
Visceral-nonvisceral disease	1.40 (0.84–2.34)	0.19	1.20 (0.61–2.36)	0.60	0.58 (0.31–1.09)	0.09	1.22 (0.57–2.62)	0.61
Bone metastasis only	1.38 (0.79–2.42)	0.26	1.15 (0.53–2.50)	0.73	2.14 (0.90–5.09)	0.09	2.59 (0.85–7.90)	0.10
Number of metastatic site	1.20 (0.71–2.03)	0.49	1.16 (0.62–2.16)	0.65	1.27 (0.69–2.35)	0.45	1.14 (0.56–2.33)	0.71
Endocrine-sensitive or -resistant disease	2.53 (1.46–4.39)	0.001	0.37 (0.20–0.67)	0.001	2.91 (1.29–6.58)	0.01	0.28 (0.12–0.66)	0.004
Dose reduction	0.88 (0.51–1.51)	0.63	0.89 (0.49–1.62)	0.70	0.65 (0.33–1.27)	0.21	1.45 (0.71–3.00)	0.31
Concurrent use of PPIs	1.06 (0.64–1.75)	0.82	0.98 (0.59–1.65)	0.95	1.36 (0.66–2.77)	0.40	1.71 (0.78–3.78)	0.18

Abbreviation: ECOG PS, Eastern Cooperative Oncology Group (ECOG) performance status; PPIs, proton pump inhibitors.

## Data Availability

Not applicable.

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
