# Peer review of "Does Proton Pump Inhibitors Decrease the Efficacy of Palbociclib and Ribociclib in Patients with Metastatic Breast Cancer?"

_medicina, 2023, doi:10.3390/medicina59030557_

Round 1

Reviewer 1 Report

Dear Author(s),

Thank you for this interesting manuscript. To the best of my knowledge, this is the first study indicating that concurrent PPI administration with palbociclib or ribociclib have no negative impact of PFS in MBC patients.

Methodology is adequate, results are clearly presented and conclusions are supported by the results. I just have several minor comments to disclose:

i) Please provide power analysis in order to precise the sample needed for valid results.

ii) Since this was a retrospective study how did you "instruct patients not to take CYP3A4 inhibitors or enhancers". Please rather say that this is a regular practice or that this was an exclusion criteria, whatever do you prefer rather.

iii) In the methods section it should also be stated that "no concurrent PPIs group" also did not take antacids nor H2 antagonists! Or mention this as a potential exclusion criteria.

iv) Consider to state that using different PPIs within "concurrent PPI group" is also a limitation, due to potentially different potency within agents.

v) How did you grade adverse events? As per CTCAE or? Please mention within the text.

Author Response

Reviewer 1 Thank you for this interesting manuscript. To the best of my knowledge, this is the first study indicating that concurrent PPI administration with palbociclib or ribociclib have no negative impact of PFS in MBC patients.

Thank you for your positive comments. 

Methodology is adequate, results are clearly presented and conclusions are supported by the results. I just have several minor comments to disclose: i) Please provide power analysis in order to precise the sample needed for valid results.

We have explained how power analysis was done in the Materials and Methods/Statistical analysis section. The following descriptions have been added to the revised manuscript: “When the “progression free survival time” is considered as the primary hypothesis to be investigated in the study, the sample size was calculated according to the expected distribution of different treatment options. When calculating at 95% level of confidence and 5% type 1 error levels, assuming a pooled standard deviation of 9 units, the study would require a sample size of 51 for each group (i.e. a total sample size of 102, assuming equal group sizes), to achieve a power of 80% and a level of significance of 5% (two sided), for detecting a true difference in means between the test and the reference group of 5 months. (Reference: Dhand, N. K., & Khatkar, M. S. (2014). Statulator: An online statistical calculator. Sample Size Calculator for Comparing Two Independent Means. Accessed 24 February 2023 at http://statulator.com/SampleSize/ss2M.html ).”

ii) Since this was a retrospective study how did you "instruct patients not to take CYP3A4 inhibitors or enhancers". Please rather say that this is a regular practice or that this was exclusion criteria, whatever do you prefer rather.

We have corrected the wrong description. The sentences have been changed as follows in the revised version: “Patients using cytochrome P450 3A4 (CYP3A4) antagonists or enhancers were not allowed to be included in the study. Additionally, patients had to receive PPIs such as omeprazole 40 mg, pantoprazole 40 mg, rabeprazole 20 mg, esomeprazole 40 mg, and lansoprazole 30 mg with breakfast in order to be eligible.”

iii) In the methods section it should also be stated that "no concurrent PPIs group" also did not take antacids nor H2 antagonists! Or mention this as a potential exclusion criterion.

We have revised the sentence to include information about antacids and H2 antagonists. The sentence has been changed as follows in the revised version:  “Patients were classified as “concurrent PPIs” if PPIs were given for at least two thirds of the palbociclib or ribociclib therapy period, and “no concurrent PPIs” if no PPIs nor antacids and H2 antagonists were given during palbociclib or ribociclib therapy. ” 

iv) Consider to state that using different PPIs within "concurrent PPI group" is also a limitation, due to potentially different potency within agents.

We have mentioned that the receipt of different PPIs can interfere with efficacy due to different potency within agents in the discussion/limitation section in the revised manuscript. 

v) How did you grade adverse events? As per CTCAE or? Please mention within the text.

We have explained the grading method of adverse events in Materials and Methods section in the revised manuscript as follows:  “Toxicity was graded according to the Common Terminology Criteria for Adverse Events (CTCAE v5).” 

Reviewer 2 Report

This investigation aimed to determine the impacts of concurrent PPIs on progression free survival (PFS) in patients with hormone receptor-positive and HER2-negative metastatic breast cancer who were being managed with palbociclib or ribociclib  as either the initial or subsequent line of therapy. 

Major concerns: 

- Conclusion might only be relevant for the PPIs they studied and not for example for omeprazole. I refer to a large study recently presented during SABCS 2022 where omeprazole affected adverse event, dose reductions and mPFS substantially

- Very heterogeneous group of 220 patients with recently treated metastatic breast cancer during a 2 years period. The group is heterogeneous and difficult to compare for median PFS, the primary endpoint in this study. The variety of PPIs they consider is also heterogeneous and might affect the study's outcome (interaction between PPI and certain drugs is PPI dependent). Most patients received the CDK 4/6i in 2nd or later lines and mPFS is affected by a lot of variables they didn't include like efficacy of the previous (endocrine therapy) line which is also important when studying mPFS in the small group of endocrine sensitive cases (9 cases in ribo group using PPI concomittantly). In the abstract they show results independent of endocrine sensitivity which is not correct; in the text thet assess palbo and ribo separately and present data for endocrine sensitive and endocrine resistant cases whereafter they present data by endocrine sensitivity whereas mPFS is not always reached (too short follow-up). 

- Again, it is crucial to know the disease free duration in the group with endocrine sensitive disease.

-  The number of variables they include in the model is too large for the small subgroups to be statistically sound

-They state 77.3% being postmenopausal but that should be 165/220 = 75%

-They state 59% having visceral disease but this is 108/220 = < 50%

-Discussion is too long and repetition with introduction

Author Response

Reviewer 2 This investigation aimed to determine the impacts of concurrent PPIs on progression free survival (PFS) in patients with hormone receptor-positive and HER2-negative metastatic breast cancer who were being managed with palbociclib or ribociclib as either the initial or subsequent line of therapy. 

Major concerns:  - Conclusion might only be relevant for the PPIs they studied and not for example for omeprazole. I refer to a large study recently presented during SABCS 2022 where omeprazole affected adverse event, dose reductions and mPFS substantially

Thank you for your suggestion. We have explained that our results may not represent all of the PPIs. In this study, omeprazole was less commonly used. The following information have been added to the limitations section with according reference. “The other limitation is related to the unequal use of PPIs in number.  Lansoprazole, pantoprazole and esomeprazole were the most commonly used PPIs in the present work. PPIs may differ within each other in terms of potency. Receipt of different PPIs has the potential to interfere differently with efficacy of the concurrently used treatments. Again, the results of the present study may not be generalized for all PPIs. Omeprazole was the least frequently used PPI in this study, which means that findings may not represent patients using omeprazole. An exploratory analysis of recent phase II PARCIFAL trial and future trials may help to assess if the results of this study can be replicated in patient populations using predominantly omeprazole. ”

- Again, it is crucial to know the disease free duration in the group with endocrine sensitive disease.

As we wanted to specify the comparison of PFS times, we didn’t mention about DFS times in the manuscript. Thirty-one patients (18 patients in palbociclib group, 13 patients in ribociclib group) had endocrine-sensitive disease when de novo metastatic patients were excluded. Median disease free survival (DFS) was 154.17 months (72.02-241.12) in palbociclib group and 120.51 months (80.49-202.58) in ribociclib group. 

-  The number of variables they include in the model is too large for the small subgroups to be statistically sound

-They state 77.3% being postmenopausal but that should be 165/220 = 75% -They state 59% having visceral disease but this is 108/220 = < 50%

We have corrected the ratios.

-Discussion is too long and repetition with introduction

We have shortened the discussion section. 

Reviewer 3 Report

Thank you for the opportunity to review this manuscript. It is an interesting article, which determines the impacts of concurrent PPIs on progression free survival in patients with hormone receptor-positive and HER2-negative metastatic breast cancer who were being managed with palbociclib or ribociclib. However, the article has some flaws which authors should justify. Main problem is the sample size. I recommend adding sample size calculation in the manuscript and discuss it. It is questionable to make a conclusion if the sample size is questionable. This makes the content insignificant. Moreover, methodology could be better described- it is not clear how was death treated/analysed. It might be good to see graphical presentation, use flow- chart to represent participants of study entered into the analysis.

Time of administration of drugs is important for the DDI during the absorption phase. IPP, when used alone are advised to be taken one hour before meal, while with CDK4/6 inhibitors with the meal. How did you assure that patients were adherent with this recommendation?

Other issues:

Abstract- methods does not include important information, such as design of the study

Language editing is needed. I noticed a lot of language problems that make parts of the text incomprehensible.

Define abbreviation when first time mentioned and be consistent throughout the text (please see PD)

Author Response

Reviewer 3 Thank you for the opportunity to review this manuscript. It is an interesting article, which determines the impacts of concurrent PPIs on progression free survival in patients with hormone receptor-positive and HER2-negative metastatic breast cancer who were being managed with palbociclib or ribociclib. However, the article has some flaws which authors should justify. Main problem is the sample size. I recommend adding sample size calculation in the manuscript and discuss it. It is questionable to make a conclusion if the sample size is questionable. This makes the content insignificant. Moreover, methodology could be better described- it is not clear how was death treated/analysed. It might be good to see graphical presentation, use flow- chart to represent participants of study entered into the analysis.

Thank you for your suggestions. We have added a flow-chart to represent participants. We have explained how power analysis was done in the Materials and Methods/Statistical analysis section. The following descriptions have been added to the revised manuscript: “When the “progression free survival time” is considered as the primary hypothesis to be investigated in the study, the sample size was calculated according to the expected distribution of different treatment options. When calculating at 95% level of confidence and 5% type 1 error levels, assuming a pooled standard deviation of 9 units, the study would require a sample size of 51 for each group (i.e. a total sample size of 102, assuming equal group sizes), to achieve a power of 80% and a level of significance of 5% (two sided), for detecting a true difference in means between the test and the reference group of 5 months. (Reference: Dhand, N. K., & Khatkar, M. S. (2014). Statulator: An online statistical calculator. Sample Size Calculator for Comparing Two Independent Means. Accessed 24 February 2023 at http://statulator.com/SampleSize/ss2M.html ).”

Time of administration of drugs is important for the DDI during the absorption phase. IPP, when used alone are advised to be taken one hour before meal, while with CDK4/6 inhibitors with the meal. How did you assure that patients were adherent with this recommendation?

We have standard suggestions in daily practice for orally used drugs. We confirmed that the patients complied with these recommendations in this study.

Other issues: Abstract- methods does not include important information, such as design of the study

We have added more information about design of the study in abstract/methods section.

Language editing is needed. I noticed a lot of language problems that make parts of the text incomprehensible.

The manuscript was checked by native speaker. 

Define abbreviation when first time mentioned and be consistent throughout the text (please see PD)

We have checked and corrected the abbreviations in the manuscript. 

Round 2

Reviewer 2 Report

The authors continue to state that PPIs given with palbo or ribo do not affect PFS. They shoud state that the most used PPI's in their study they used in the paper didn't affect PFS.

Determing PFS in a retrospective study is tricky as a lot of variables can't be controlled for; this is a limitation when interpreting these data

The discussion is too long

Author Response

Q1-The authors continue to state that PPIs given with palbo or ribo do not affect PFS. They shoud state that the most used PPI's in their study they used in the paper didn't affect PFS.

A1-Thank you for your valuable comments. We have explained that our data represents the most commonly used PPIs (lansoprazole, pantoprazole and esomeprazole) in our study and corrected the relevant sentences in Abstract, and Discussion sections in the revised manuscript.

Q2-Determing PFS in a retrospective study is tricky as a lot of variables can't be controlled for; this is a limitation when interpreting these data.

A2-We have added data about PFS in Discussion/limitations section in the revised manuscript.

Q3-The discussion is too long.

A3-We have further shortened Discussion section in the revised manuscript. 

Reviewer 3 Report

Thank you for the revised version of the manuscript. Incorporated changes have improved the manuscript and I can recommend it to be accepted.

Author Response

Q1-Thank you for the revised version of the manuscript. Incorporated changes have improved the manuscript and I can recommend it to be accepted.

A1-Thank you for your valuable comments.